# Clinical Profiles in Renal Patients with COVID-19

**DOI:** 10.3390/jcm9082665

**Published:** 2020-08-18

**Authors:** María Dolores Arenas, Marta Crespo, María José Pérez-Sáez, Silvia Collado, Dolores Redondo-Pachón, Laura Llinàs-Mallol, María Milagro Montero, Judith Villar-García, Carlos Arias-Cabrales, Francesc Barbosa, Anna Buxeda, Carla Burballa, Laia Sans, Susana Vázquez, Anna Oliveras, Marisa Mir, Sara Outón, Isabel Galcerán, Eulalia Solá, Adriana Sierra, Clara Barrios, Eva Rodríguez, Higini Cao, Roberto Güerri-Fernández, Juan Pablo Horcajada, Julio Pascual

**Affiliations:** 1Department of Nephrology, Hospital del Mar, 08003 Barcelona, Spain; 96685@psmar.cat (M.C.); 60665@psmar.cat (M.J.P.-S.); scollado@psmar.cat (S.C.); 61110@psmar.cat (D.R.-P.); lllinas@psmar.cat (L.L.-M.); 60559@psmar.cat (C.A.-C.); fbarbosa@psmar.cat (F.B.); abuxeda@psmar.cat (A.B.); 61079@psmar.cat (C.B.); lsans@psmar.cat (L.S.); svazquez@psmar.cat (S.V.); 87052@psmar.cat (A.O.); mmir@psmar.cat (M.M.); sara.outon@gmail.com (S.O.); bel.galceran@gmail.com (I.G.); 63567@psmar.cat (E.S.); asierra@psmar.cat (A.S.); 94489@psmar.cat (C.B.); erodriguez@psmar.cat (E.R.); hcao@psmar.cat (H.C.); 2Department of Infectious Diseases, Hospital del Mar, 08003 Barcelona, Spain; mmontero@psmar.cat (M.M.M.); jvillar@psmar.cat (J.V.-G.); rguerri@psmar.cat (R.G.-F.); Jhorcajada@psmar.cat (J.P.H.)

**Keywords:** COVID-19, SARS-CoV-2, RT-PCR, kidney transplantation, hemodialysis, pneumonia

## Abstract

The COVID-19 pandemic has led to frequent referrals to the emergency department on suspicion of this infection in maintenance hemodialysis (MHD) and kidney transplant (KT) patients. We aimed to describe their clinical features comparing confirmed and suspected non-confirmed COVID-19 cases during the Spanish epidemic peak. Confirmed COVID-19 ((+)COVID-19) corresponds to patient with positive RT-PCR SARS-CoV-2 assay. Non-confirmed COVID-19 ((−)COVID-19) corresponds to patients with negative RT-PCR. COVID-19 was suspected in 61 patients (40/803 KT (4.9%), 21/220 MHD (9.5%)). Prevalence of (+)COVID-19 was 3.2% in KT and 3.6% in MHD patients. Thirty-four (26 KT and 8 MHD) were (+)COVID-19 and 27 (14 KT and 13 MHD) (−)COVID-19. In comparison with (−)COVID-19 patients, (+)COVID-19 showed higher frequency of typical viral symptoms (cough, dyspnea, asthenia and myalgias), pneumonia (88.2% vs. 14.3%) and LDH and CRP while lower phosphate levels, need of hospital admission (100% vs. 63%), use of non-invasive mechanical ventilation (36% vs. 11%) and mortality (38% vs. 0%) (*p* < 0.001). Time from symptoms onset to admission was longer in patients who finally died than in survivors (8.5 vs. 3.8, *p* = 0.007). In KT and MHD patients, (+)COVID-19 shows more clinical severity than suspected non-confirmed cases. Prompt RT-PCR is mandatory to confirm COVID-19 diagnosis.

## 1. Introduction

COVID-19 has spread throughout the world, affecting more than 4.5 million persons since late December 2019 [1,2]. Variable incidences have been described in different countries depending on the severity of the epidemic and the testing policy. In Spain, most diagnoses have been obtained in symptomatic persons, thus the number of confirmed cases has increased from 4209 cases by 13 March 2020 [3] to more than 230,000 confirmed cases and 27,650 deaths by early May [4].

Symptoms of COVID-19 are not specific: variable combinations of fever, cough, dyspnea, myalgia and asthenia comprise the usual syndrome at presentation [5,6]. Consequently, clinical presentation usually leads to suspicion, which needs confirmation with adequate tests. Early identification of patients infected with COVID-19 has important implications for isolation, reduction of the risk of transmission and management. Polymerase chain reaction (PCR) tests for SARS-CoV-2 identify viral ribonucleic acid (RNA) in respiratory tract, and it is considered the “gold standard” for acute infection. As a new disease, information on the clinical presentation and early evolution is not comprehensive, and studies comparing clinical and analytical presentation and final outcomes between suspected cases with PCR positive and negative are unavailable.

Patients with renal diseases, including the whole spectrum from kidney transplant (KT) to maintenance hemodialysis (MHD) patients, are at high risk for infection and pneumonia, both bacterial and viral [7,8]. The available reports on COVID-19 in these populations underline important incidences and poor outcomes, so early and efficient identification and diagnosis is of utmost importance [9,10,11].

Our study paper to present the clinical profiles and outcomes of confirmed and non-confirmed cases of renal patients with suspected COVID-19 at our institution and establishes the differing profile and outcome between both patient groups.

## 2. Materials and Methods

### 2.1. Patients and COVID-19 Diagnosis

An observational study was planned in all KT recipients and MHD patients who were studied in the Hospital del Mar between 12 March 2020 and 21 April 2020 for suspected COVID-19 infection. Any patient admitted during that 40-day period of pandemic peak with COVID-19 compatible signs was considered as suspected. In particular, fever was the main symptom leading to potential COVID-19.

Demographic, clinical, laboratory and radiological data were retrospectively obtained through review of medical records to complete a prospectively established patient list, launched at the beginning of the epidemic. Each patient was followed until death or at least up to 14 days of follow-up. Variables included were age, sex, race, vascular access disposition, comorbidities, baseline treatments, clinical symptoms, oxygen saturation at admission, chest *X*-ray findings, laboratory parameters, specific treatments for COVID-19 and outcomes.

Resolution was defined as disappearance of symptoms and discharge.

### 2.2. SARS-CoV-2 Testing

Testing for presence of the SARS-CoV-2 virus was undertaken using qualitative reverse transcription PCR (RT-PCR). Confirmed COVID-19 ((+)COVID-19) corresponds to a patient with positive PCR SARS-CoV-2 assay of a specimen collected on a nasopharyngeal swab or bronchoalveolar lavage. Non-confirmed COVID-19 ((−)COVID-19) corresponds to a patient with negative RT-PCR. Several molecular diagnostic platforms based on real-time RT-PCR assay were performed, including Roche cobas^®^ and Abbott^®^ Real-time SARS-CoV2 Assay^®^ (Chicago, IL, USA).

The LightMix^®^ Modular SARS-CoV-2 assays (TIB Molbiol, Berlin, Germany) were performed on a LightCycler^®^ 480 II system (Roche Diagnostics). The screening reaction detects the viral E-gene using a region conserved in SARS, SARS-CoV-2 and other bat-related coronaviruses. Following a positive E-gene detection, the confirmation of SARS-CoV-2 was performed by detecting the specific RdRP gene. According to the manufacturer, limit of detection was 15 copies per reaction when performing 40 cycles. The Abbott RealTime SARS-CoV-2 assay is a dual target assay for the detection of the viral RdRP and N genes by means of target-specific probes. This assay was performed on the Abbott m2000 system. According to the manufacturer, the limit of detection was 100 virus copies/mL, which corresponded to the lowest concentration level with positive rates >95%. According to the package insert, the clinical performance evaluation study showed both 100% positive (95% CI 94, 100) and negative (95% CI 88.8, 100) percent agreement. In a subsequent study, the sensibility and specificity of this assay were found to be 93% and 100%, respectively [12].

The study was performed under the principles of the Declaration of Helsinki and was approved by the hospital ethics committee.

### 2.3. Statistical Analysis

Quantitative variables with a normal distribution are expressed as mean and standard deviation (SD). Categorical variables are summarized as counts and percentages. Categorical (Chi-square and Fisher’s exact test) variables were used as appropriate. Mann–Whitney U test was used to determine if the means of groups are significantly different from each other. A logistic regression multivariate model was performed to adjust factors associated with confirmed COVID-19. A *p* < 0.05 was considered statistically significant. Statistical analysis was performed using SPSS V. 21.0 (SPSS Inc., Chicago, IL, USA).

## 3. Results

### 3.1. Patients

COVID-19 was suspected in 61 renal patients (40 KT and 21 MHD). The prevalence of suspected cases was 4.9% in KT recipients (40 out of 803 actively followed KT) and 9.5% in MHD patients (21 out of 220 MHD patients) at Hospital del Mar and our two external facilities.

Finally, 34 patients resulted to be RT-PCR SARS-CoV-2 positive ((+)COVID-19) and 27 patients RT-PCR SARS-CoV-2 negative ((−)COVID-19). Eight of 21 MHD patients (38%) and 26 of 40 KT patients (65%) were (+)COVID-19 (*p* = 0.041). Prevalence of (+)COVID-19 patients was 3.2% in KT patients and 3.6% in MHD patients.

Absence of COVID-19 in the 27 (−)COVID-19 patients was confirmed in 17 cases with repeated and consecutive negative PCR: two (*n* = 9), three (*n* = 3) or four (*n* = 5) tests. The mean number of attempts in the unconfirmed group was 2.1 ± 1.2 determinations. All (+)COVID-19 except one, who required two PCR determinations, were positive on the first attempt.

Final diagnoses were other infections (*n* = 17: gastroenteritis (*n* = 3); intravenous catheter (*n* = 3); urinary tract (*n* = 3); diabetic foot (*n* = 2); and endocarditis, psoas abscess, subcutaneous cell tissue, influenza B, pericarditis and other pneumonia (*n* = 1 each)), heart failure (*n* = 5), ischemic colitis (*n* = 1) and unknown (*n* = 4).

SARS-CoV-2 PCR testing was also performed in all MHD patients (*n* = 220) in Hospital del Mar and the two external MHD facilities dependent on the hospital and no asymptomatic cases were detected.

### 3.2. Comparison between COVID-19 Confirmed and Non-Confirmed Cases

Demographic factors, comorbidities and baseline treatments in COVID-19 confirmed and non-confirmed cases are shown in Table 1. Patients with (+)COVID-19 were older and more obese than (−)COVID-19 patients, although the differences did not reach statistical significance. No other differences were found between both groups.

Clinical characteristics and laboratory findings at admission are summarized in Table 2.

The three major symptoms observed in (+)COVID-19 patients were fever (88.2%), cough (79.4%) and dyspnea (67.6%). Other common symptoms included asthenia and myalgia. Infrequent symptoms included headache, diarrhea, ageusia and anosmia. Fever, cough, dyspnea, asthenia and myalgia were less frequent in suspected but (−)COVID-19 compared with (+)COVID-19 patients. Pneumonia was present in 88.2% of (+)COVID-19 patients and only in 14.3% of (−)COVID-19 ones. This pneumonia was bilateral in 76.5% of (+)COVID-19 patients and 11.1% of (−)COVID-19 ones (*p* < 0.001). C-reactive protein (CRP) and lactic dehydrogenase (LDH) were significantly higher in (+)COVID-19 patients. Patients with (+)COVID showed a tendency to show a decreased number of lymphocytes (*p* = 0.07) and a lower transferrin saturation (*p* = 0.061).

As expected, (+)COVID-19 patients more frequently received specific drug treatments than non-confirmed ones (Table 3).

Hydroxychloroquine, azithromycin, ceftriaxone and steroids were the most frequently prescribed drugs. Lopinavir/ritonavir and tocilizumab were used in some confirmed cases but never in non-confirmed ones. Prophylactic enoxaparin was prescribed in 44% of confirmed cases and in 11% of non-confirmed ones. Two thirds of suspected but (−)COVID-19 patients did not receive any specific treatment.

All confirmed cases underwent hospital admission, in comparison with only 63% of non-confirmed ones (Table 3). Intensive care unit (ICU) admission rate was lower than expected for the severity of the disease, and only 7 out of 34 (+)COVID-19 patients were given intensive care, with some additional ones requiring noninvasive mechanical ventilation (36.4%). These therapies were even less frequent in (−)COVID-19 cases (11.1%) (*p* < 0.01), although endotracheal intubation was not significantly different between both groups.

Acute kidney injury was more frequent in KT patients with (+)COVID-19 than in those with (−)COVID-19, although the difference was not statistically significant.

At the end of follow-up, 13 (38%) (+)COVID-19 patients had died, while no (−)COVID-19 patient had died, and all of them were safely discharged (*p* < 0.001). Mortality in (+)COVID was associated with lower lymphocyte levels (0.60 (0.52) vs. 1.15 (0.8); *p*: 0.041), bilateral pneumonia (*p*: 0.006) and treatment with azithromycin (*p*: 0.028) (Table 4).

The elapsed time for the different events is summarized in Table 3. Mean time from the onset of symptoms to admission was similar in confirmed and suspected non-confirmed patients (5.8 vs. 5.2 days), but longer in (+)COVID patients who finally died than in survivors (8.5 vs. 3.8, *p* = 0.007). Time from admission to resolution and discharge was significantly longer in (+)COVID-19 patients (12.4 days) than in non-confirmed ones (5 days).

### 3.3. Comparison of Pneumonia due to COVID-19 and Other Pneumonias

Overall, 35 patients were diagnosed as having pneumonia, 30 (+)COVID-19 and 5 (−)COVID-19. Patients with (+)COVID-19 pneumonia had more frequently cough and asthenia than patients with other pneumonias. Confirmed COVID-19 pneumonia was more frequently bilateral (83.3% vs. 40%) and the expert radiologist informed compatible chest *X*-ray in 29 out of the 30 cases, compared with only one out of five (−)COVID-19 patients with pneumonia. No statistical differences in laboratory parameters were found. Mortality of (+)COVID-19 pneumonia was significantly higher (43.3% vs. 0%) than in other pneumonias.

A multivariate logistic regression model confirmed that pneumonia was the unique factor significantly associated with (+)COVID-19 diagnosis before PCR positivity (Table 5).

### 3.4. Comparison of Confirmed vs. Suspected COVID-19 in Kidney Transplant Recipients

Confirmed COVID-19 KT patients were older than (−)COVID-19 ones (Table 5). No other significant differences were found between both groups in terms of demographic factors, comorbidities and baseline treatments. The three major symptoms observed in (+)COVID-19 KT patients were fever, cough and dyspnea. Only one confirmed KT patient referred ageusia and none anosmia. Fever, cough, dyspnea, asthenia and myalgia were significantly less frequent in (−)COVID-19 compared with (+)COVID-19 KT patients. (+)COVID-19 KT patients showed a higher frequency of pneumonia (88.5%), mostly bilateral, compared with (−)COVID-19 (*p* < 0.001).

Contact with a healthcare facility in the month prior to infection was frequent in both groups, although more frequent in confirmed cases (84% vs. 46.7%, *p* = 0.008).

Laboratory findings at admission showed non-significant differences between confirmed and non-confirmed KT cases except for LDH, which was significantly higher in (+)COVID-19 KT patients (*p* < 0.001). (+)COVID-19 KT patients more frequently received drug treatments than non-confirmed cases. Seventy-three percent of (−)COVID-19 KT patients did not receive any specific COVID-19 treatment, and, in the (+)COVID-19 KT patient group, only one did not receive it.

Hospital admission was 100% of (+)COVID-19 KT patients and 60% of non-confirmed ones (*p* < 0.001). ICU admission and endotracheal intubation were not significantly different between the groups. At the end of follow-up, fatality rate in (+)COVID-19 KT patients was 42%, while no non-confirmed cases died (*p* < 0.001).

Mean time from onset of symptoms to admission was 6.7 and 3 days in confirmed and non-confirmed KT patients, respectively.

### 3.5. Comparison of Confirmed vs. Suspected COVID-19 in Hemodialysis Patients

No significant differences were found between (+)COVID-19 MHD patients and non-confirmed ones regarding demographic factors, comorbidities and baseline treatments (Table 5). Regarding clinical presentation at admission, (+)COVID-19 and (−)COVID-19 MHD patients showed similar symptoms (fever, cough and dyspnea) and only asthenia was more frequent in (+)COVID-19 MHD patients than in (−)COVID-19 ones. No MHD patient with (+)COVID-19 referred headache or myalgia. Most (+)COVID-19 MHD patients showed bilateral pneumonia compared with (−)COVID-19 ones.

Confirmed COVID-19 MHD patients had lower levels of hemoglobin and lymphocytes compared with non-confirmed ones, but without statistical differences. Serum phosphate was significantly lower and fibrinogen was significantly higher in (+)COVID-19 MHD patients. D-dimer levels were similar in (+)COVID-19 and (−)COVID-19 groups.

With respect to specific COVID-19 treatments, confirmed MHD patients received similar drugs as confirmed KT patients. Only two confirmed MHD patients did not receive any specific COVID-19 treatment compared with 60% of non-confirmed ones.

All MHD patients with (+)COVID-19 were admitted to the hospital, in comparison with only 60% of non-confirmed cases (*p* < 0.05). ICU admission and endotracheal intubation were similar in confirmed and non-confirmed MHD patients, but noninvasive mechanical ventilation was significantly more used in (+)COVID-19 MHD patients (57.1%) than in non-confirmed ones (13.3%) (*p* = 0.03). At the end of follow-up, three of eight (+)COVID-19 MHD patients had died (37.5%), while no (−)COVID-19 patients died. Time from admission to resolution and discharge was significantly shorter in (−)COVID-19 MHD cases (23 vs. 4.7 days, *p* = 0.006).

## 4. Discussion

COVID-19 was suspected in 61 renal patients seen in the emergency department of our hospital during the first month of the Spanish COVID-19 pandemic, and it was confirmed through RT-PCR in two-thirds of them. Prevalence of confirmed COVID-19 was 3.2% in KT patients and 3.6% in MHD patients. This prevalence in MHD patients was lower than the 10–41% reported by other groups [10,11,13,14]. This low prevalence is relevant, as Barcelona has been one of the places with the highest prevalence of COVID-19 in Spain [4], and Catalonia region ranks second in number of cases included in the Spanish COVID-19 register (18% out of 868 patients), behind Madrid (36%) [15]. Our group promptly designed a protocol to limit the spread of the infection in its dialysis facilities that probably has influenced this low prevalence [16]. Regarding KT, (+)COVID-19 recipients were older than (−)COVID-19 ones, and the incidence reported is in line with other published data [17,18,19,20,21]. Cardiovascular disease, diabetes, chronic respiratory diseases, high blood pressure and cancer are the main risk factors described in COVID-19 [22] Nevertheless, in our population, no differences in pre-existing comorbidities were observed between COVID-19 confirmed and non-confirmed renal patients except that confirmed cases were most frequently obese. A worse prognostic has been described in young obese people [23]. The absence of differences with respect to the general population may be partly because the prevalence of these comorbidities is already high in renal patients or due to the small sample size.

Recent reports suggest a milder clinical presentation of COVID-19 in renal patients compared with general population [10,11,14,24,25]. However, our COVID-19 confirmed patients presented a high rate of early clinical symptoms, much more intense than those patients whose RT-PCR was negative. The most frequent symptoms observed were fever, cough and dyspnea [6]. Ageusia and anosmia were infrequent but only appeared in confirmed cases, and we consider them as pathognomonic for the disease. Olfactory and gustatory dysfunction have been described as common initial symptoms of infection in the general population showing prevalence of 52.7% and 43.9%, respectively [26]. Asymptomatic presentation in renal patients is frequent, reaching up to 40% [10,24,25] and has been attributed to the immune system dysfunction in MHD patients [24]. Interestingly, none of our COVID-19 confirmed cases were asymptomatic.

Chest *X*-ray was highly suggestive of COVID infection in most patients with RT-PCR positive (KT and MHD) being rare in non-confirmed ones. All images were examined by expert radiologists who issued a report on the degree of suspected infection. The typical radiological pattern shows multifocal patchy peripheral consolidations in bilateral lungs, except for left upper lung zone [27,28]. In our experience, most confirmed patients presented pneumonia, which was bilateral in most cases, while it was infrequent in non-confirmed ones. These data are consistent with other published data [11,14,24,25,29]. In fact, the presence of pneumonia was the unique factor associated with true COVID-19 before PCR confirmed positivity in our series. Normal chest *X*-ray and overlapping of imaging features between COVID-19 and other viral pneumonias have been described in COVID-19 patients [27]. We were able to compare characteristics of COVID-19 related pneumonia and other pneumonias in suspected but non-confirmed COVID-19 patients in our cohort. Severity and mortality were much higher in COVID-19 pneumonia. In contrast, Zhao et al. [28] showed no differences in symptoms between COVID-19 and non-COVID-19 pneumonias, and fever and cough were the most common symptoms found. Conversely, we found that cough and asthenia were more frequent in COVID-19 pneumonia than in non-COVID-19 ones. Bilateral affectation is more frequent in COVID-19 related pneumonia [28,30]. Chest CT scan has been suggested as a useful tool to screen the suspected cases of COVID-19 infection and discard false negative cases [30,31]. Interestingly, our radiologists classified most (+)COVID-19 pneumonias as compatible while only one case of (−)COVID-19 pneumonia was considered. Chest TC was performed in only one PCR negative patient: images were suggestive, he was treated as a COVID-19 patient and he turned PCR positive later.

COVID-19 distinctive biochemical alterations described in early reports from China [28,32,33,34] were comparable to our findings. Elevation of CRP, LDH, D-dimer and lymphopenia were all frequently observed but only CRP and LDH were significantly higher in confirmed COVID-19 patients as compared with non-confirmed ones. Serum creatinine and serum phosphate were significantly lower in confirmed COVID-19 patients. These low phosphate levels may reflect a higher degree of malnutrition. Further studies are needed to assess hypophosphatemia as a potential risk factor for COVID-19.

Confirmed COVID-19 were treated more frequently than non-confirmed ones, following hospital protocols. They received hydroxychloroquine plus azithromycin, as the first-line treatment, and lopinavir/ritonavir or tocilizumab, as second line. These treatments are similar to those used in other hospitals [11,13,35]. Only one non-confirmed patient received hydroxychloroquine and azithromycin due to a high level of suspicion, while no non-confirmed patients received lopinavir/ritonavir or tocilizumab. Steroids [36] and prophylactic anticoagulation with low-molecular-weight heparin [37] have been recommended in COVID-19 infection and these treatments were more frequently used in confirmed patients. Only one thrombotic and no hemorrhagic events were observed in COVID-19 confirmed patients. A number of our patients, especially MHD ones, received vitamin D, as it may reduce COVID-19 severity through several mechanisms [38].

Although some earlier series suggested a good prognosis of COVID-19 in renal patients [38,39], recent reports show a worse outcome [10,11,12,13,17,18,40] and a higher mortality rate (16–40%) than that observed in the general population [41,42]. Our study confirms these findings. All confirmed cases underwent hospital admission, in comparison with only 58% of non-confirmed cases. Outcomes of (+)COVID-19 were worse than non-confirmed patients after hospitalization. Probability of noninvasive mechanical ventilation, ICU admission, endotracheal intubation and death was higher in (+)COVID than in non-confirmed patients. This could be explained by the fact that the sensitivity of the test is greater in more viremic patients [43,44], which is usually related to more severe clinical conditions [45]. The unique factor related with mortality in our study was the time from onset of symptoms to admission. Median days from onset or identification of symptoms until admission was 8.5 in non-survivors and 3.8 in survivors. Early identification of suspected COVID-19 is important, not only for isolation and decreasing the risk of transmission, but also for prompt admission and action.

This study has limitations. The small sample size and the short median follow-up may have had an impact on the generalizability of our analyses. Center bias cannot be ruled out. First, due to the retrospective nature, some laboratory tests such as IL-6, fibrinogen and serum ferritin were not done in all patients. Similarly, symptoms and analytical evolution were not collected in all cases. In addition, suboptimal technique to take the swab sample cannot be ruled out in all cases. However, the systematic inclusion of all suspected cases gives rigorous clinical value and supports the notion that unspecific symptoms should lead to prompt PCR to confirm or discard this severe disease.

## 5. Conclusions

Despite confirmed and non-confirmed COVID-19 cases showed symptoms compatible with COVID-19 at admission, confirmed COVID-19 showed more severe clinical outcome and analytical alterations at admission and at early follow-up. Mortality was earlier and higher in renal patients with COVID-19, but it was very low in initially suspected but non-confirmed cases.

It is essential to perform early RT-PCR to confirm diagnosis in suspected COVID-19 cases in KT or MHD patients to improve prognosis. Suspected non-confirmed cases usually show a benign course and iatrogenic therapeutic measures should be especially avoided.

## Figures and Tables

**Table 1 jcm-09-02665-t001:** Baseline characteristics of COVID-19 confirmed and non-confirmed cases.

	COVID-19 Confirmed (*n* = 34)	COVID-19 Non-Confirmed (*n* = 27)	*p*
**Demographics and profile**			
Age (years. mean ± Standard deviation (SD))	69 ± 10.1	62.1 ± 17	0.056
Age > 65 years (*n*, %)	25 (73.5)	13 (48.1)	0.042
Male Sex (*n*, %)	24 (70.6%)	21 (77.8%)	0.279
Caucasian race (*n*, %)	27 (79.4%)	23 (85.1%)	0.519
Kidney transplantation/Hemodialysis (*n*)	26/8	14 / 13	0.041
**Comorbidities and treatments**			
Catheter as vascular access	1 (2.9%)	2 (7.4%)	0.413
Current Smoking status (*n*, %)	6 (17.6%)	8 (29.6%)	0.136
Obesity (*n*, %)	12 (35.3%)	4 (14.8%)	0.064
Diabetes mellitus (*n*, %)	13 (38.2%)	13 (50%)	0.424
Arterial hypertension (*n*. %)	31 (91.2%)	25 (92.6%)	0.602
Previous cancer (*n*, %)	9 (26.5%)	4 (13.3%)	0.216
Heart disease (*n*, %)	15 (44.1%)	12 (44.1%)	0.592
Lung disease (*n*, %)	11 (32.4%)	9 (33.3%)	0.575
Dyslipidemia	20 (58.8%)	20 (74.1%)	0.308
ACE inhibitors/Anti AT1-receptor blockers (*n*, %)	8 (23.6%)	8 (29.6%)	0.476
Vitamin D derivatives (*n*, %)	6 (17.6%)	9 (33.3%)	0.224
Cinacalcet/Etecalcetide (*n*, %)	1 (2.9%)	4 (14.8%)	0.112

**Table 2 jcm-09-02665-t002:** Clinical and analytical profile of COVID-19 confirmed and non-confirmed cases.

	COVID-19 Confirmed (*n* = 34)	COVID-19 Non-Confirmed (*n* = 27)	*p*
**Clinical symptoms**			
Fever	30 (88.2%)	21 (77.8%)	0.227
Cough	27 (79.4%)	12 (44.4%)	0.005
Dyspnea	23 (67.6%)	9 (33.3%)	0.008
Asthenia	19 (55.9%)	5 (18.5%)	0.001
Myalgia	13 (38.2%)	3 (11.1%)	0.016
Diarrhea	9 (26.5%)	6 (22.2%)	0.469
Headache	4 (11.8%)	3 (11.1%)	0.630
Ageusia	2 (5.9%)	0 (0%)	0.200
Anosmia	1 (2.9%)	0 (0%)	0.369
**Lung disease**			
Pneumonia	30 (88.2%)	5 (14.3%)	0.000
Bilateral Pneumonia	26 (76.5%)	3 (11.1%)	0.000
**Laboratory findings**			
Hemoglobin (g/dL, mean ± SD)	10.8 (2.1)	11.5 (2.1)	0.228
White blood cells (×10^3^/μL, mean ± SD)	8.2 (3.9)	9.8 (5.4)	0.239
Lymphocytes (×10^3^/μL, mean ± SD)	0.96 (0.74)	1.36 (0.88)	0.070
Neutrophils (×10^3^/μL, mean ± SD)	6.79 (3.65)	6.97 (3.87)	0.964
Platelets (×10^3^/μL, mean ± SD)	144 (69)	169 (99)	0.225
D-dimer (mcg/L, median (IQR))	940 (290–34,742)	1180 (190–6080)	0.391
Fibrinogen (UI/L, mean ± SD)	836.5 (271)	651(325)	0.241
AST (UI/L, mean ± SD)	29.1 (26.4)	19 (14.7)	0.172
LDH (UI/L, mean ± SD)	378 (263.5)	180 (94.8)	0.009
LDH > 230 UI/L * (*n*, %)	23 (76.7)	3 (21.4)	0.001
CPK (U/L, mean ± SD)	62.3 (47.8)	78.8 (66)	0.406
C-reactive protein (mg/L, mean ± SD)	12.1 (8.9)	6.5 (7.9)	0.017
C-reactive protein > 4 mg/L * (*n*, %)	25 (73.5)	11 (45.8)	0.032
Ferritin (ng/mL, median (IQR))	1450 (66–6987)	1172 (224–2860)	0.173
Phosphate (mg/dL; mean ± SD)	2.33 (1.96)	4.50 (1.65)	0.002

* ROC curve cut-offs.

**Table 3 jcm-09-02665-t003:** COVID-19 treatments and outcomes in confirmed and non-confirmed cases.

	COVID-19 Confirmed (*n* = 34)	COVID-19 Non-Confirmed (*n* = 27)	*p*
**Treatments**			
Hydroxychloroquine (*n*, %)	31 (93.9%)	4 (15.4%)	0.000
Azithromycin (*n*, %)	29 (87.9%)	8 (30.8%)	0.000
Ceftriaxone (*n*, %)	20 (62.5%)	4 (15.4%)	0.001
Steroids (*n*, %)	21 (63.6%)	1 (3.7%)	0.000
Enoxaparin (*n*, %)	15 (44.1%)	3 (11.1%)	0.005
Lopinavir/ritonavir (*n*, %)	7 (20.6%)	0 (0%)	0.027
Tocilizumab (*n*, %)	8 (23.5%)	0 (0%)	0.006
Vitamin D (*n*, %)	4 (11.8%)	0 (0%)	0.065
Nothing (*n*, %)	3 (8.8%)	18 (66.7%)	0.000
**Outcomes**			
Hospital admission (*n*, %)	34 (100%)	17 (63%)	0.000
ICU admission (*n*, %)	7 (21.2%)	4 (14.8%)	0.406
Non-invasive mechanical ventilation (*n*, %)	12 (36.4%)	3 (11.1%)	0.028
Endotracheal intubation (*n*, %)	6 (18.2%)	3 (11.1%)	0.367
AKI in non-dialysis patients (*n*, %)	9/26 (34.6%)	3/15 (20%)	0.119
Idem requiring dialysis (*n*, %)	5/26 (19.2%)	3/15 (20%)	0.262
Thrombosis (Pulmonary thromboembolism) **	1 (3%)	0 (0%)	0.557
Resolution (*n*, %)	19 (55.9%)	27 (100%)	0.000
Death (*n*, %)	13 (38.2%)	0 (0%)	0.000
**Time * from onset of symptoms and:**			
-RT-PCR positive or negative	4.9 (0–15)	5.5 (0–43)	0.847
-Hospital admission	5.8 (0–18)	5.2 (0–43)	0.641
-Intensive care unit admission	0.5 (0–2)	0.6 (0–2)	0.910
**Time * from Hospital admission and:**			
-Acute kidney injury	0 (0–0)	0.2 (0–2)	0.817
-Resolution	12.4 (1–23)	5.0 (0–33)	0.010
-Death	7.5 (0–16)	-	-

* Time expressed as median (IQR) days. ** KT patient.

**Table 4 jcm-09-02665-t004:** Multivariate logistic regression analysis of factors associated with (+)COVID-19.

	OR	IC95%	*p*
**Demographics and profile**			
Age > 65 years old	4.32	0.49–38.98	0.19
Hemodialysis vs. Kidney transplantation	3.04	0.39–23.44	0.29
Obesity	4.51	0.31–66.27	0.27
**Clinical features**			
Pneumonia	11.64	1.29–104.69	0.028
**Serum laboratory parameters**			
LDH (UI/L)	1.012	0.99–1.03	0.10
C-reactive protein (mg/L)	1.006	0.86–1.17	0.93

**Table 5 jcm-09-02665-t005:** Confirmed and non-confirmed COVID-19 in kidney transplant recipients and hemodialysis patients.

	Kidney Transplantation			Hemodialysis		
	COVID-19 Confirmed (*n* = 26)	COVID-19 Non-Confirmed (*n* = 14)	*p*	COVID-19 Confirmed (*n* = 8)	COVID-19 Non-Confirmed (*n* = 13)	*p*
**Baseline characteristics**						
Age (years, mean ± Standard deviation (SD))	70.3 (9.1)	59.2 (16.1)	0.010	64.7 (12,8)	65.3 (16.9)	0.936
Time on KT/HD (years. median (IQR))	6.1 (6.9)	3.9 (3.3)	0.241	2.5 (2.1)	4.1 (3.6)	0.306
Male Sex (*n*, %)	17 (65.4%)	10 (71.4%)	0.395	7 (87.5%)	11 (84.6%)	0.684
Contact with healthcare facilities prior month	22 (84.6%)	5 (35.7%)	0.013	3 (37.5%)	7 (53.8%)	0.392
Known infected contact	1 (3.8%)	5 (35%)	0.014	3 (37.5%)	3 (23.1%)	0.410
**Clinical profile and signs at admission**						
Fever	24 (92.3%)	10 (71.4%)	0.099	6 (75%)	11 (84.6%)	0.498
Cough	22 (84.6%)	6 (42.9%)	0.019	5 (71.4%)	6 (46.2%)	0.201
Dyspnea	19 (73.1%)	6 (42.9%)	0.062	4 (50%)	3 (23.1%)	0.213
Asthenia	13 (50%)	3 (21.4%)	0.048	6 (75%)	2 (15.4%)	0.011
Myalgia	13 (50%)	2 (14.3%)	0.027	0 (0%)	1 (7.7%)	0.619
Diarrhea	7 (26.9%)	3 (21.4%)	0.508	2 (25%)	3 (23.1%)	0.656
Headache	4 (15.4%)	1 (7.1%)	0.418	0 (0%)	2 (15.4%)	0.371
Ageusia	1 (3.8%)	0 (0%)	0.650	1 (14.3%)	0 (0%)	0.381
Anosmia	0 (0%)	0 (0%)	-	1 (14.3%)	0 (0%)	0.381
Pneumonia	23 (88.5%)	4 (7.1%)	0.000	7 (87.5%)	1 (7.7%)	0.001
Bilateral pneumonia	19 (73.1%)	2 (14.3%)	0.000	6 (75%)	1 (7.7%)	0.003
**Laboratory findings at admission**						
Hemoglobin (gr/dL, mean (SD))	11.0 (2.2)	11.0 (1.7)	0.993	10 (1.4)	11.9 (2.5)	0.081
White blood cells (×10^3^/μL, mean (SD)	8.46 (4.0)	11.3 (6.1)	0.097	8.3 (4.3)	8.5 (4.5)	0.912
Lymphocytes (×10^3^/μL, mean (SD))	1.0 (0.82)	1.1 (0.84)	0.818	0.73 (0.36)	1.6 (0.93)	0.034
Neutrophils (×10^3^/uL, mean (SD))	7.0 (3.78)	7.5 (4.23)	0.510	6.8 (3.4)	6.0 (3.6)	0.635
Platelets (×10^3^/μL, mean (SD))	142 (66)	199 (122)	0.055	151 (84)	139 (60)	0.711
Serum Creatinine (mg/dL, mean (SD))	2.4 (1.3)	3.8 (2.7)	0.047	5.7 (5.2)	6.5 (3.19)	0.672
**D-dimer (mcg/L, median (IQR))**	860 (290–34,732)	1410 (550–2780)	0.505	1210 (840–15,100)	1570 (190–6080)	0.449
Fibrinogen (UI/L, mean (SD))	647 (416–1000)	1000 (1000)	0.239	1000 (0)	540 (320)	0.021
AST (UI/L, mean (SD))	29.6 (28.1)	11.8 (2.9)	0.082	22 (6.3)	24.7 (17.5)	0.741
LDH (UI/L, mean (SD))	378 (259)	137 (107)	0.023	260 (92)	218 (65)	0.325
IL-6 (ng/mL, mean (SD))	214 (372)	21.7 (8)	0.612	120 (77)	98.6	0.810
C-reactive protein (mg/L, mean (SD))	12.6 (8.0)	8.5 (8.8)	0.161	8.0 (11.2)	6.4 (6.9)	0.403
Ferritin (ng/mL, median (IQR))	1500 (66–4873)	672 (534–2397)	0.203	868 (534–2397)	931 (224–2860)	0.630
**COVID-19 treatments**						
Hydroxychloroquine (*n*, %)	23 (92%)	3 (23%)	0.000	7 (87.5%)	1 (7.7%)	0.001
Ceftriaxone (*n*, %)	15 (62.5%)	2 (15%)	0.007	4 (50%)	2 (15.4%)	0.115
Azitromicin (*n*, %)	22 (88%)	4 (30%)	0.001	6 (75%)	4 (30.8%)	0.049
Lopinovir/ritovir (*n*, %)	5 (19.2%)	0 (0%)	0.090	1 (12.5%)	0 (0%)	0.381
Tocilizumab (*n*, %)	6 (23.1%)	0 (0%)	0.051	2 (25%)	0 (0%)	0.058
Steroids (*n*, %)	14 (56%)	1 (7.1%)	0.003	6 (75%)	0 (0%)	0.001
Enoxaparin (*n*, %)	8 (30.8%)	2 (14.3%)	0.193	7(87.5%)	1 (7.7%)	0.001
Nothing (*n*, %)	1 (3.8%)	10 (71%)	0.000	1 (12.5%)	8 (61.5%)	0.119
**Outcomes and evolution**						
Hospital admission (*n*, %)	26 (100%)	9 (64%)	0.003	8 (100%)	8 (61.5%)	0.044
ICU admission (*n*, %)	6 (23.1%)	2 (14%)	0.412	1 (12.5%)	2 (15.4%)	0.684
Noninvasive mechanical ventilation (*n*, %)	8 (30.8%)	1 (7.1%)	0.091	4 (50%)	2 (15.4%)	0.088
Endotracheal intubation (*n*, %)	5 (19.2%)	2 (14%)	0.529	1 (12.5%)	1 (7.7%)	0.629
Resolution (*n*, %)	14 (53.8%)	14 (100)	0.000	5 (62.5%)	13 (100%)	0.017
Death (*n*, %)	10 (38.5%)	0 (0%)	0.006	3 (37.5%)	0 (0%)	0.042
**Times and evolution of events (days, mean (IQR))**						
Time from symptoms onset to RT-PCR	5.6 (0–15)	4.1 (0–23)	0.449	2.3 (0–10)	6.8 (0–43)	0.430
Time from symptoms onset to admission	6.7 (0–18)	3.0 (0–14)	0.062	5.8 (0–18)	7.1 (0–43)	0.460
Time from admission to AKI	0 (0–0)	0.12 (0–2)	0.817	-	-	-
Time from admission to resolution	11.2 (1–21)	4.8 (0–33)	0.066	23 (23)	5.3 (0–12)	0.008
Time from admission to death	7.2 (0–16)	0	0.001	8.0 (0–15)	0 (0-0)	0.423

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
