# Peer review of "Clinical Profiles in Renal Patients with COVID-19"

_jcm, 2020, doi:10.3390/jcm9082665_

Round 1

Reviewer 1 Report

The authors describe the clinical and laboratory features, treatment, and outcome data for 61 out of 1023 patients actively followed by the nephrology service at a single health care system.  34 of the patients were PCR positive for COVID-19, and 27 patients were hospitalized contemporaneously with non-COVID illnesses.  All variables are compared between COVID and non-COVID patients, further stratified for patients with transplants and patients on dialysis.  I have concerns about the overall statistical power of the study and the subset analysis, as well as the joint analysis of dialysis and transplant patients.

Major critiques

  1. Page 5, line 99 and Table 2. Is a significance level of 0.05 appropriate here, or should the p value take into account the multiple hypothesis testing? Please provide information on the power level of your study in the methods section.
  2. Results, page 5, lines 108-112. The authors describe that COVID-19 was excluded by repeated negative PCR tests. Please elaborate further on the need for testing multiple times.  Were any patients tested multiple times with negative results before converting to a positive result, or did all positive patients test positive on the first attempt?
  3. Table 4 is inappropriate, as the comparison is underpowered, with only 5 pneumonias in the non-COVID group.  Please remove this table.  Perhaps a more qualitative approach to the radiographic findings would be informative here. Were there features on imaging common in cases of COVID-19 pneumonia (aside from the bilateral involvement), such as lobar consolidation, crazy-paving pattern, pleural effusion, etc?
  4. Table 6 is essentially redundant with Tables 2 and 3. Lumping together the kidney transplant and dialysis patients is an artificial construct. The two groups of patients have little in common save for their care by nephrologists and frequent interactions with the health care system.  One group is immunosuppressed with intact kidney function and the other has a generally intact immune system with marked kidney dysfunction.  Table 6 clearly shows that there are differences in the presentations, laboratory findings at admission, and outcomes between transplant and dialysis groups.  Please justify presenting the data for transplant recipients and dialysis patients together before separating the groups.
  5. In the study, 13 patients with COVID died. Were there significant differences in the presentations or treatments received in this group?

Minor critiques:

Introduction, Page 2, line 40 and table 5. Consistency of number formatting (commas vs. periods)

Methods, Page 4, line 96. “variables” should perhaps be “tests”.  Consider revising that sentence.

Table 6. Confirm the percentage on Transplant nonconfirmed pneumonia 4 (7.1%)

Results, page 14, line 213-214: Revise the sentence beginning “Only one of…” for clarity.

Author Response

Reviewer 1

1.- Page 5, line 99 and Table 2. Is a significance level of 0.05 appropriate here, or should the p value take into account the multiple hypothesis testing? Please provide information on the power level of your study in the methods section.

We used the conventional level of significance at p<0.05, and this is provided in statistical Methods section.

2.- Results, page 5, lines 108-112. The authors describe that COVID-19 was excluded by repeated negative PCR tests. Please elaborate further on the need for testing multiple times.  Were any patients tested multiple times with negative results before converting to a positive result, or did all positive patients test positive on the first attempt?

We have clarified this in the text. See line 114-116.  “Absence of COVID-19 in the 27 nCOVID-19 patients was confirmed in 17 cases with repeated and consecutive negative PCR: two (n=9), three (n=3) or four (n=5)” The mean number of attempts in the unconfirmed group was 2.1 (1.2) determinations. All COVID-19 confirmed except one, who required 2 PCR determinations, were positive on the first attempt.

3.- Table 4 is inappropriate, as the comparison is underpowered, with only 5 pneumonias in the non-COVID group.  Please remove this table.  Perhaps a more qualitative approach to the radiographic findings would be informative here. Were there features on imaging common in cases of COVID-19 pneumonia (aside from the bilateral involvement), such as lobar consolidation, crazy-paving pattern, pleural effusion, etc

 We agree with the reviewer and have removed the table and summarized the main results in the text. The only main difference between both pneumonia subgroups was the bilateral involvement in (+)COVID cases.

4.- Table 6 is essentially redundant with Tables 2 and 3. Lumping together the kidney transplant and dialysis patients is an artificial construct. The two groups of patients have little in common save for their care by nephrologists and frequent interactions with the health care system.  One group is immunosuppressed with intact kidney function and the other has a generally intact immune system with marked kidney dysfunction.  Table 6 clearly shows that there are differences in the presentations, laboratory findings at admission, and outcomes between transplant and dialysis groups.  Please justify presenting the data for transplant recipients and dialysis patients together before separating the groups.

Our study aims to present the clinical profiles and outcomes of confirmed and non-confirmed cases of renal patients (KT and MHD) with suspected COVID-19. Both kind of patients are attended in the emergency room by the same nephrologists. Both groups are immunosuppressed, the KT patients by immunosuppressive drug treatment and the MHD ones by their profund chronic kidney disease itself. We tried to establish the different profile and outcome between (+) and (-) COVID 19  to help nephrologist identify both types of patients. It is essential to perform early RT-PCR to confirm diagnosis in suspected COVID-19 cases in KT recipients or MHD patients to improve prognosis. Suspected non-confirmed cases usually show a benign course and iatrogenic therapeutic measures should be especially avoided in both group of patients. Knowing the differences between both groups, of course, we also assessed both populations separately.

5.- In the study, 13 patients with COVID died. Were there significant differences in the presentations or treatments received in this group?

Regrettably, no very relevant differences were detected at presentation between those patients who died or survived. And the treatment response was very limited. See line 189-191. “Mortality in + COVID was associated with lower lymphocyte levels (0.60 (0.52) vs. 1.15 (0.8); p: 0.041), bilateral pneumonia ((P: 0.006) and treatment with azithromycin (p: 0.028)”.

Minor critiques:

Introduction, Page 2, line 40 and table 5. Consistency of number formatting (commas vs. periods)

We  have changed it

Methods, Page 4, line 96. “variables” should perhaps be “tests”.  Consider revising that sentence.

 We agree with you. We  have changed it

Table 6. Confirm the percentage on Transplant nonconfirmed pneumonia 4 (7.1%)

 Yes. It is wrong. It is 28%

Results, page 14, line 213-214: Revise the sentence beginning “Only one of…” for clarity

We have changed it . Seventy-three percent of (-)COVID-19 KT patients did not received any specific COVID-19 treatment, and in  (+)COVID-19 KT patient group , only one did not received  it.

Reviewer 2 Report

The research has analyzed the clinical characteristics of patients with renal disease from a single center during the pandemic.
The research is relevant because acute kidney injury is commonly encountered in COVID patients because of the abundance of ACE2 receptors in the kidney.
This is an original retrospective investigation from a single center. The COVID pneumonia was compared with other pneumonias, symptoms were compared between confirmed and negative patients in renal patients. The risk factors, clinical characteristics, laboratory data and treatment were analyzed. Tables are well presented. The paper is well written and easy to read. The conclusions consistent with the research aimed to address the severity of the disease in confirmed patients.

Author Response

Thank you for your comments.

Reviewer 3 Report

The authors elegantly describe their vast experience with and very relevant observations on kidney transplant recipients and dialysis patients in the setting of COVID-19.

The manuscript is reader-friendly, written in great English.

The tables are especially helpful.

I have only a few minor comments:

  • The findings may be easier to follow if the groups were classified as +COVID-19 and -COVID-19 rather than cCOVID-19 and nCOVID-19
  • What is the manufacturer reported/estimated incidence of false + and of false - for the PCR testing?
  • Were all patients evaluated in the ER (both admitted and not admitted to the hospital) tested for SARS-CoV-2 infection?
  • Did treatment with steroids improve outcomes in cCOVID-19 cases?
  • Do the authors think that a greater number of patients would have resulted in a statistically significant difference with respect to other co-morbidities in addition to obesity?

Author Response

Reviewer 3

1.- The findings may be easier to follow if the groups were classified as +COVID-19 and -COVID-19 rather than cCOVID-19 and nCOVID-19.

Thanks for this comment. We have changed it in the text

2.- What is the manufacturer reported/estimated incidence of false + and of false - for the PCR testing?

We have added an explanation from the manufacturers:

The LightMix® Modular SARS-CoV-2 assays (TIB Molbiol, Berlin, Germany) are performed on a LightCycler® 480 II system (Roche Diagnostics). The screening reaction detects the viral E-gene using a region conserved in SARS, SARS-CoV-2 and other bat-related coronaviruses. Following a positive E-gene detection, the confirmation of SARS-CoV-2 is performed by detecting the specific RdRP gene. According to the manufacturer, limit of detection was 15 copies per reaction when performing 40 cycles. The Abbott RealTime SARS-CoV-2 assay is a dual target assay for the detection of the viral RdRP and N genes by means of target-specific probes. This assay is performed on the Abbott m2000 system. According to the manufacturer, the limit of detection was 100 virus copies/mL, which corresponded to the lowest concentration level with positive rates >95%. According to the package insert, the clinical performance evaluation study showed both 100% positive (95% CI 94, 100) and negative (95% CI 88.8, 100) percent agreement. In a subsequent study, the sensibility and specificity of this assay were found to be 93% and 100%, respectively (Degly-Angeli et al., Validation and verification of the Abbott RealTime SARS-CoV-2 assay analytical and clinical performance. J Clin Virol. 2020.)

3.- Were all patients evaluated in the ER (both admitted and not admitted to the hospital) tested for SARS-CoV-2 infection?

Yes, all patients evaluated were tested for SARS-CoV-2 infection

4.- Did treatment with steroids improve outcomes in cCOVID-19 cases?

In +COVID group, corticosteroid treatment was administered to 20 patients, 6 of them died and 14 survived. Similarly, in the group that did not receive corticosteroids, 6 died and 7 survived. There were no statistically significant differences in terms of mortality in the group that received corticosteroids. (p:0.282)

5.- Do the authors think that a greater number of patients would have resulted in a statistically significant difference with respect to other co-morbidities in addition to obesity?

A greater number of patients would have resulted in a statistically significant difference with respect to other co-morbidities in addition to obesity, however comorbidities such as diabetes, hypertension, or dyslipidemia are frequent in the renal population and prevalence of risk factors was similar in (+)COVID and (–)COVID.  In renal patients, classical cardiovascular risk factors in the general population could lose prognostic power in renal population. Other studies (Goicoechea M, Sánchez Cámara LA, Macías N, et al. COVID-19: clinical course and outcomes of 36 hemodialysis patients in Spain. Kidney Int. 2020;98(1):27-34. doi:10.1016/j.kint.2020.04.031) have found that none of the classical cardiovascular risk factors in the general population were associated with higher mortality.